# The Role of Globalization, Economic Growth and Natural Resources on the Ecological Footprint in Thailand: Evidence from Nonlinear Causal Estimations

Hafezali Iqbal Hussain [1,2], Muhammad Haseeb [1], Fakarudin Kamarudin [3,4,*], Zdzisława Dacko-Pikiewicz [5] and Katarzyna Szczepańska-Woszczyna [5]

1 Taylor's Business School, Taylor's University Lakeside Campus, 1 Jalan Taylors, Subang Jaya 47500, Malaysia; hafezali.iqbalhussain@taylors.edu.my (H.I.H.); muhammad.haseeb@taylors.edu.my (M.H.)
2 University of Economics and Human Sciences in Warsaw, Okopowa 59, 01-043 Warsaw, Poland
3 School of Business and Economics, Universiti Putra Malaysia, Serdang 43400, Malaysia
4 EIS-UPMCS Centre for Future Labour Market Studies, SOCSO, Putrajaya 62100, Malaysia
5 Department of Management, Faculty of Applied Sciences, WSB University, 41-300 Dabrowa Górnicza, Poland; zdacko@wsb.edu.pl (Z.D.-P.); kszczepanska@wsb.edu.pl (K.S.-W.)
* Correspondence: fakarudin@upm.edu.my

**Abstract:** The environmental issue has become a global problem that needs to be examined frequently, motivating researchers to investigate it. Thus, the present study has investigated the asymmetric impact of globalization, economic growth and natural resources on the ecological footprint in the presence of environmental Kuznets curve (EKC) in Thailand. The study has used annual time series data from 1970 to 2018. The study applied a novel method of nonlinear autoregressive distributive lag (ARDL). In particular, the current study has investigated the effect of positive and negative shocks on the independent variable on the dependent variable. The findings have confirmed that the effect of globalization and natural resources are significant and nonlinear. However, the effect of negative shocks of globalization and natural resources is more dominant on the ecological footprint in Thailand than the positive shocks of both variables. Moreover, the present study has also tested the presence of EKC in Thailand, and the findings confirm the presence of an inverted U-shape curve in the Thailand economy.

**Keywords:** globalization; economic growth; natural resources; ecological footprint; environmental Kuznets curve; Thailand

## 1. Introduction

The degradation of the environment is one of the most urgent challenges facing the global community. Resource utilization at higher rates could impact the environment. With this backdrop of decreasing resources, climate change is seen as one of the major challenges of the modern human race. It is safe to mention that everyone is responsible for this progressive worsening of living conditions regardless of any division of developed or developing countries. It is a widely known fact that natural resources are the assets of every nation, enabling countries to be preferred when it comes to trade. Natural resource prevalence and environmental issues are not issues that are limited to geography; rather, it is a global challenge [1]. Using an economic lens, Reference [2] sees natural resources as the key to the progress of a country, describing natural resources as "factors of production provided by nature, which is, soils, forests, grassland, air, water, minerals, fuels, etc." In this vein, it is argued that the increasing depletion of natural resources is a severe threat to sustainable development.

Numerous methods are used to examine the impact of human and economic activities on environmental degradation. One such method is the ecological footprint, which first came to the surface in 1990, when it was described as "use of land and water for production

of all resources consumed by humans and for eliminating the waste material generated by the population" [3]. In this process of production and consumption, the concept is generally used to examine the environmental situation, which is the outcome of these activities, and was earlier measured through $CO_2$ emissions [4]. Currently, the ecological footprint is generally used as an evaluation measure for environmental degradation [4]. A large part of the use of ecological footprints could be best described by the notion that an elaborative and comprehensive method is required to examine the effect of human activities on the environment. It is argued that natural resources positively affect ecological footprint and enable a country to manage its problem. For instance, in climate change or extreme heat, forestry significantly absorbs tons of carbon from spreading into the atmosphere. Further, trees reduce temperature and improve rainfall in the long run, which can help deal with water scarcity [3]. However, the excessive use of these natural resources, which is an "irreversible process", can pose grave challenges to human society and the prospect of environmental sustainability [5].

Apart from this, enhanced pressure on the ecological footprint results from greater demand for consumption and usage involved in attaining economic advancements, trade expansion, globalization, etc. With the increase in countries' desire to become highly globalized, the supply–demand tug of war has pushed countries to work together to minimize the supply–demand gap. Following this concept, many studies believe that globalization contributes to increasing pressure on the environment [6,7]. While explaining why globalization contributes to increasing pressure on the environment, they borrowed the idea of "race to bottom" earlier used by [8], which means that when host countries look for foreign direct investment, they relax their environmental regulations. Such relaxation generally allows countries to shift those businesses, which results in environmental challenges. Moreover, Reference [9] conducted a study on the relationship of globalization with the ecological footprint and found it to be a stressor, but the situation is different for the social aspect of globalization. They believed that the more the societies are interconnected and aware, the less the chances exist that stressors can play their role. On the other hand, the alternative view asserts that the emergence of globalization can have positive and negative effects on environmental changes [10]. In this regard, it is argued that developing countries often benefit from the learning curve of developed countries who have already honed their skills in confronting environmental challenges by developing green technology and processes.

Among the developing countries, the current use of natural resources in China is exceptional as the country alone has utilized 50% of global coal resources, and the impacts are quite visible in the forms of pollution and extreme weather situations [11]. Despite their importance, it has been observed that minimal work had been conducted in this specific area [3]. In this regard, Reference [12] shed light on natural resources in the Middle East and North Africa (MENA) region and drew attention to environmental and water shortage challenges. By describing the worsening situation of the region, Reference [12] used facts cited in work by the World Bank [13] and mentioned that the MENA region is one of the "poorest regions of the world in terms of renewable water and arable land", which is why the region is facing severe pollution challenges, environmental degradation and above all water shortage. Until recently, the situation has remained the same, and problems have further worsened despite the region's richness with other natural resources. Moreover, Reference [14] discussed this perspective differently; they argue that natural resource depletion can severally impact the environment. They further added that this can be managed if renewable energy resources are used instead of non-renewable resources. This can minimize ecological footprints and help maintain the natural resources that help deal with the climate change. In a somewhat similar manner, Reference [15] indicated the impact of natural resource (extraction) on environmental degradation. They suggested that in order to fulfill the increasing demand, the extraction of natural resources at higher rates decreases the bio capacity of the environment, which eventually results in ecological footprints.

Linking the use of resources with environmental burden, Reference [16] found the case of Thailand to be alarming. The ecological condition of Thailand was in surplus until the end of 1980s. Since 1990, the biocapacity surplus of Thailand has started to decline every year, indicating a continuous decline in environmental quality. In the year 2016, the ecological footprint of the country stood at 2.5 global hectares (gha). On the other hand, the biocapacity was 1.2. Hence, the biocapacity deficit was −1.3. In this regard, Reference [17] asserted that the higher pressure on natural resources has enhanced the ecological burden. Precisely, it is believed that the extension of oil palm and rubber plantations in the country has resulted in a rising ecological footprint in energy, forest and cropland. This increase in the area where the ecological footprint was previously greater than the biocapacity has amplified the stress on the utilization of resource use in the region. Given the increase in the globalization potential of Thailand, there is a need to identify the role of globalization in influencing environmental degradation. The association is crucial due to the rise in county's economic globalization and higher dependence on trade and foreign investments [18]. Additionally, recently, the study of [15] argued that economic growth in Thailand showed a decline in environmental degradation primarily but with subsequent deterioration due to the usage of outdated technologies and increased energy intensities.

In light of the above, it is wise to assert that economic growth, globalization and natural resources preserve the critical relationship with environmental degradation. Increasing human demands is creating stress, which is more than what the contemporary ecosystem can offer. Against this backdrop, the efforts to achieve equilibrium between demand and supply factors (economic growth, globalization, and natural resources) impact environment quality (ecological footprints). Though a plethora of studies available, it is clear that grounds are available to conduct research on the nexus of these three factors on environmental degradation. Critical review shows that limited research was conducted on natural research as a mechanism to minimize ecological footprints. Other grounds are also available to research ecological aspects of globalization. Keeping in view such possibilities, the present study examines the asymmetric impact of economic growth, globalization and natural resources on the ecological footprint. The present study utilizes a novel approach, nonlinear autoregressive distributed lag (NARDL) that was introduced by [19]. In particular, the present research investigates how positive and negative shocks of globalization and natural resources affect the ecological footprint in Thailand's economy. The outcomes of this study will provide a strong understanding of the relationship of globalization and natural resources with an ecological footprint in the context of Thailand.

Several investigations in the prevailing literature have emphasized the rising environmental challenges [20,21], mostly due to increasing globalization and industrial expansion [22–24]. The progress in country's development is desirable as it provides the government with the opportunity to serve the needs of the country better and sustain their future survival [25–27]. However, over time, with higher urbanization, trade and enhanced globalization, there is a continuous rise in environmental degradation resulting in increased consumption, water, air and land pollution [28,29], and enhanced exploitation of natural resources [30]. The traditional resource curse concept demonstrates that countries with abundant natural resources experience slower economic growth rates than countries that contain limited natural resources [31]. This is due to the negative effect that is transferred to other industries that subsequently pay for the prosperity of resource-associated industries [32]. Moreover, many natural resources, such as minerals and fuels as well as fishery and forestry, have also experienced the depletion that threatens the notion of sustainable development [3,33]. Additionally, there exist several adverse environmental impacts of natural resource extractions and consumption, especially fuels, such as coal, natural gas and petroleum, which have enhanced the levels of greenhouse gases into the atmosphere, thereby causes global warming [34].

Agreeing to the continuously growing notion of environmental deterioration, Reference [35] briefed on the major cause of this challenge, which in their opinion is the global

$CO_2$ emissions that are at times attributed to the economic growth. Contrary to this, there are voices that see the critical role of economic growth in a healthier environment. For instance, Reference [36] suggested that economic growth is the objective of any country, but it should be pursued by ensuring minimal environmental damage. Similarly, studies such as [4,12] also found a similar relationship between income and ecological footprint. This kind of relationship was also seen in the countries that are part of the Belt and Road initiative, where such a relationship is visible [37].

Contrary to this, there are studies that fail to find any such relationship. By way of illustration, Reference [38] work revealed that foreign direct investment (FDI), represented as a significant part of globalization, has no valid association with the ecological footprint. Similarly, the work of [39] is worthy of mention in this regard, which suggested that the positive or negative relationship between the income levels and ecological footprints is yet to reach a decisive point. The study also argued that such a relationship is absent or minimal in higher-income countries. On the basis of this, it is safe to mention that recently the biggest contributions to the environmental degradation are the developing countries. Furthermore, studies attempted to show the relationship between early phases of growth and environmental degradation. In this regard, one of the notable examples was provided by [40], who presented the case of Sub-Saharan countries and the consumption of energy. They revealed that countries like Botswana used less electricity before 2000 than today, so this consumption was raised by 3.8% at the Sub-Saharan African level. They believe the roots of this increasing consumption (mostly by non-renewable) could be traced from the improvement of gross domestic product (GDP) growth from 2.2% to 4.9%, gradually, from 2000 to 2017. In line with this, Reference [41] proposed that every country must take into consideration the need for balance between economic growth and environmental degradation. They believe that this can be achieved by the possibilities which push developed countries to control their revenue growth and also the developing countries to control their spread. The other possibility is through the domain of EKC link. It is believed that the increased levels of economic growth and globalization put pressure on manufacturing and consumption levels, leading to demand overshoot and an increase of the ecological footprint [42,43]. Similar concerns were traditionally raised in the environmental Kuznets curve (EKC) that suggests that rising income levels deteriorate initially but ultimately improved environmental quality [44–46]. However, there also exist concerns that the EKC only exists in the footprints of country's production and does not consider the globalization components, and is not reflected from import footprints [47]. This suggests that rich countries can easily improve their ecological footprint at the cost of poor countries' environment.

Hence, recognizing the potential threats of natural resource utilization, globalization and economic growth, many studies empirically analyzed the combined and specific impact of these variables on the environment. Among them, Reference [15] examined the impact of economic growth, financial advancements and energy utilization on environmental quality. To fulfill the objective, the authors utilized the measure of ecological footprints to identify environmental degradation in eleven newly industrialized nations from 1977 to 2013. The study stated that the role of economic progress is complementary to environmental degradation and sustainable development. Overall, the results suggested that energy consumption enhanced the ecological footprint in seven of the eleven economies. Likewise, financial development also decreases the ecological footprint of China and Malaysia but increases ecological overshoot in Singapore. As for economic growth, the outcomes found mixed results. For the economies of South Africa, Philippines, Mexico and Singapore, the results confirmed the existence of an inverted U-shaped EKC curve, suggesting that a rise in income initially amplified degradation but ultimately reduced it. On the other hand, the findings of Thailand, India, Turkey, China and South Korea suggested that a rise in economic growth declined ecological footprint primarily but subsequently deteriorated the environment due to the usage of outdated technologies and increased energy intensities.

Likewise, for a panel of MENA nations, Reference [12] examined the role of output growth in environmental degradation. Using the data of fifteen MENA economies from 1975 to 2007, the authors distributed the studied nations to oil-exporting and non-oil-exporting nations. The empirical investigation findings indicated an inverted U-shaped link between economic growth and ecological footprint in the MENA economies that export oil. For the case of non-oil-exporting nations, the study found the existence of a U-shaped EKC curve, suggesting that growth led to the reduction of ecological footprint, followed by a subsequent increase. Moreover, Reference [48] also analyzed the link between economic progress and environmental degradation in Qatar between 1980 and 2011. For this, the study adopted two major proxies of environmental degradation, i.e., carbon dioxide and ecological footprint. The outcomes found that the increase in economic growth decreased $CO_2$ and ultimately deteriorated environmental quality by increasing emission levels. On the other hand, the rise in economic growth degraded the environment by enhancing the ecological footprint but eventually improved environmental conditions with reduced pressure on the ecological footprint of Qatar.

In another study, Reference [49] analyzed European economies to study the link between economic growth and environmental degradation measured by ecological footprint. For this, the study used the data of fifteen European nations from 1980 to 2013. The study results found that an increase in economic growth reduced ecological footprint but ultimately amplified it. Moreover, in a mixed panel of 116 economies, Reference [47] also investigated the role of income levels in influencing the environment. Using the measure of ecological footprint to recognize environmental degradation, the study analyzed the validity of the EKC curve from 2004 to 2008. The study's findings reported that the existence of inverted U-curve association only existed in income and domestic production links. As for import footprint, the authors found that an increase in import led to enhanced ecological footprint monotonically. Similar to [49], Reference [50] also studied the role of economic growth and ecological footprint in Europe utilizing a panel of sixteen European economies between 1997 and 2014. The outcomes of the empirical results documented that a unit increase in economic growth is likely to raise the ecological footprint by 0.81%.

Likewise, Reference [51] also analyzed the growth-environment nexus in fourteen Asian economies using ecological footprint as an indicator of environmental degradation. The study results documented the validity of the EKC curve only in the economies of Nepal, Pakistan, India and Malaysia. However, for the rest of the Asian countries, including Thailand, the study found a significant positive relationship between growth and ecological footprint. Moreover, including natural resources in the environment-growth link, Reference [3] also examined the connection between output, natural resources and environmental degradation by adopting the proxy of ecological footprint to indicate climate downfall in Pakistan. Similar to [48], the study validated the presence of an inverted U-shaped EKC curve. As for natural resources, the results reported that natural resources increase the pressure on the environment by increasing the country's ecological footprint.

In another examination of natural resources and environment connection, Reference [52] analyzed the impact of natural resources in influencing the ecological footprint of China. Evaluating the data from 1980 to 2010, the results suggested the significant role of natural resources in enhancing ecological pressure in the Chinese economy. The outcomes reported that a rise in natural resource consumption carried a negative impact on ecological footprint leading to the enhanced ecological deficit by 66 times from 1983 to 2010. Likewise, in Thailand, Reference [17] investigated the impact of palm oil and rubber industries in affecting ecological footprint of the country. The results of the study found the significant role of the studied industries in affecting the ecological footprint of Thailand. The authors suggested that in order to attain the objectives of sustainability in rubber and palm oil industries, there remained the need to involve ecological footprint figures as the crucial indicator of sustainable growth. Focusing on oil resource, Reference [53] examined the oil and ecological footprint relationship in ten OPEC economies from 1977 to 2008. The results found the significant EKC link in the economies of Iraq, Nigeria, Kuwait, Algeria,

Venezuela and Qatar and reported the presence of an inverted U-Shaped association. Furthermore, the study found that an increase in oil consumption increased the ecological footprint in the considered economies.

Studying the link between globalization and environment, Reference [6] examined the connection between ecological footprint and king of fighters (KOF) index of globalization. In doing so, the authors evaluated the panel data of 171 economies for four diverse measures of ecological footprint, i.e., consumption, production, export and import footprints. The outcomes of the investigation reported the significant link of globalization on three measures of ecological footprint. Precisely, it is found that KOF index enhances environmental degradation by increasing import, export and consumption footprints in the studied economies. Moreover, Reference [54] also examined the impact of globalization on environmental degradation by utilizing the measure of ecological footprint. For this, the study gathered the data of 146 economies from 1981 to 2009. The findings supported the significant effect of overall globalization on the ecological footprint of export and import in the panel estimation. Additionally, the results indicated that the rise in social globalization decreases the footprints of production and consumption. On the other hand, an increase in social globalization is found to have a positive relationship with the footprints of export and import. As for economic globalization, the results found that economic globalization increases all types of ecological footprints. Lastly, the study failed to find the significant association between political globalization and the measures of ecological footprints.

Assessing the role of globalization in the context of EKC link, Reference [55] analyzed the association of economic growth and KOF index of globalization with an ecological footprint in South Asian economies from 1975 to 2017. The findings of the study validated the presence of EKC curve in the studied economies by reporting an inverted U-shaped association between economic growth and ecological footprint. Moreover, globalization tends to degrade environmental condition by enhancing the ecological footprint in South Asian countries. Moreover, distinguishing the environmental degradation into two measures of carbon and ecological footprint, Reference [10] investigated the influence of globalization on the Malaysian environment. In doing so, the authors used the data from 1971 to 2014 and reported the significant link of globalization in enhancing the carbon footprint of Malaysia. On the other hand, the study found that globalization persisted with an insignificant impact on the ecological footprint of Malaysia. In another recent study, Reference [42] also analyzed the impact of globalization on the ecological footprint of fifteen globalized economies from 1970 to 2017. They applied the innovative method of quantile-on-quantile regression, and the study reported the significant impact of globalization on the ecological footprint of the studied economies.

In addition, a study by Sharif et al. Reference [42] suggested that the ecological footprint has gained greater importance with time and needs to be investigated with some economic factors. Moreover, palm oil and rubber industries in Thailand have a greater effect on the ecological footprint and have a more significant impact on the country's economy. Thus, to fulfill these gaps and to consider the importance of this area, it motivates the researchers to examine the role of economic factors on ecological footprint. Thus, this research aims to examine the asymmetric effect of NAR and globalization (GLO) on the ecological footprint (EFP). Moreover, another aim is to test the environmental Kuznets curve in the Thailand economy.

## 2. Materials and Methods

This research examines the asymmetric effect of NAR and GLO on EFP. Moreover, it also tests the environmental Kuznets curve in the Thailand economy. Therefore, following this statement, the following equation is used for empirical estimation:

$$\ln \text{EFP}_t = f(\ln \text{GDP}_t, \ln \text{GDP}_t^2, \ln \text{NAR}_t, \ln \text{GLO}_t,) \qquad (1)$$

EFP = ecological footprint;
GDP = gross domestic product;

NAR = natural resources;
GLO = globalization.

The linear form of the above equation is as under:

$$\ln \text{EFP}_t = \beta + \beta_1 \ln \text{GDP}_t + \beta_2 \ln \text{GDP}_t^2 + \beta_3 \ln \text{NAR}_t + \beta_4 \ln \text{GLO}_t + \mu_t \quad (2)$$

lnEFP = logarithm of ecological footprint;
lnGDP = logarithm of gross domestic product;
lnNAR = logarithm of natural resources;
lnGLO = logarithm of globalization.

In econometric modeling, several approaches like ARDL, ECM or Granger causality are used to find the relationship between variables. The multiple regression analysis was applied since the independent variables are varied [56–61]. These approaches are generally used when the relationship between two or more variables needs to be checked, especially for the long run. One of the salient features of these approaches is their ability to take into account the asymmetric nature of the data. The data were collected from world development indicators and KOF index for globalization from 1970 to 2018. Contrary to this linear regression model is used for checking the linear relationship among variables, although they cannot check the variable nonlinear behavior. Taking the work on ARDL framework to a higher level, which is asymmetric ARDL co-integration approach: [19] based their efforts on the early contributions of [62,63], i.e., initial form ARDL framework. This newly developed approach captures short-term disturbances and also any asymmetries. This study is focused on exploring any such asymmetric effects of the independent variable on the dependent variable.

$$\text{EFP}_t = \alpha_0 + \alpha_1 \text{GDP}_t + \alpha_2 \text{GDP}_t^2 + \alpha_3 \text{NAR}_t^+ + \alpha_4 \text{NAR}_t^- + \alpha_5 \text{GLO}_t^+ + \alpha_6 \text{GLO}_t^- + \varepsilon_t \quad (3)$$

In this equation, the ecological footprint is denoted by EFP, whereas natural resources are represented with NAR, whilst globalization is denoted by GLO. Moreover, GDP and GDP$^2$ represent the gross domestic product and square of it, whereas the co-integrating vectors will be estimated by $\alpha$ (ranging from $\alpha 1$, $\alpha 2$, $\alpha 3$ to $\alpha 6$ in the equation). In addition, the partial positive and negative effect of focus variables (natural resources and globalization) on ecological footprint is also incorporated in Equation (3).

Considering Equation (2), as proposed by [19], the extended asymmetric ARDL model is shown as follows:

$$\Delta \text{EFP}_t = \beta_0 + \beta_1 \text{EFP}_{t-1} + \beta_2 \text{GDP}_{t-1} + \beta_3 \text{GDP}_{t-1}^2 + \beta_4 \text{NAR}_{t-1}^+ + \beta_5 \text{NAR}_{t-1}^- + \beta_6 \text{GLO}_{t-1}^+ + \beta_7 \text{GLO}_{t-1}^- +$$
$$\sum_{i=1}^{m} \delta_{1i} \Delta \text{EFP}_{t-1} + \sum_{i=0}^{n} \delta_{2i} \Delta \text{GDP}_{t-i} + \sum_{i=0}^{n} \delta_{3i} \Delta \text{GDP}_{t-i}^2 + \sum_{i=0}^{p} \delta_{4i} \Delta \text{NAR}_{t-i}^- + \sum_{i=0}^{p} \delta_{5i} \Delta \text{NAR}_{t-i}^+ + \quad (4)$$
$$\sum_{i=0}^{q} \delta_{6i} \Delta \text{GLO}_{t-i}^+ + \sum_{i=0}^{r} \delta_{7i} \Delta \text{GLO}_{t-i}^- + u_i$$

Equation (4) includes several lag orders, denoted by m, n, p, q and r. Similarly, NAR and GLO related effect of the disturbance, whether negative or positive on the EFP are reflected by β1, β2, β3, β4, and β5. Apart from them, Equation (4) also considers short term effects, which are represented by $\sum_{i=0}^{n} \delta 2i$, $\sum_{i=0}^{n} \delta 3i$, $\sum_{i=0}^{n} \delta 4i$, and $\sum_{i=0}^{n} \delta 5i$, respectively. Further, this is worth mentioning that a nonlinear long association among variables can also be examined using the NARDL approach.

The asymmetric ARDL model follows several steps; for instance, several tests like Augmented Dicky–Fuller and Phillips–Perron are performed as a first step. These tests will detail the stationarity of the variables, although it is not required when ARDL model is used. Researchers like [64–66] agree with the notion and argue that stationarity in a variable only acts as a hindrance if 1(2) series is present; otherwise, series such as 1(0), 1(1), or their mixture pose no threat to the application of ARDL model. In the backdrop of such a potential challenge, it is sane to examine these series for valid findings as the second step ordinary least square method is applied for the estimation of Equation (8). Aligned to this,

the method of [67] was used for following SIC information criterion and general to specific approach in this regard. Lastly, co-integration was evaluated through the bound test, so the asymmetric ARDL model was used. This step made it a possibility to derive an asymmetric cumulative dynamic multiplier effect of percentage change in $NAR_{t-1}^{+}$, $NAR_{t-1}^{-}$, $GLO_{t-1}^{+}$, $GLO_{t-1}^{-}$, accordingly as shown as follows:

$$s_h^{+}(NAR) = \sum_{j=0}^{h} \frac{\partial EFP_{t+i}}{\partial NAR_{t-1}^{+}} \tag{5}$$

$$s_h^{-}(NAR) = \sum_{j=0}^{h} \frac{\partial EFP_{t+i}}{\partial NAR_{t-1}^{-}} \tag{6}$$

$$s_h^{+}(GLO) = \sum_{j=0}^{h} \frac{\partial EFP_{t+i}}{\partial GLO_{t-1}^{+}} \tag{7}$$

$$s_h^{-}(GLO) = \sum_{j=0}^{h} \frac{\partial EFP_{t+i}}{\partial GLO_{t-1}^{-}} \tag{8}$$

## 3. Results

In the initial step, the present research applied fundamental statistics, which is called descriptive statistics. The findings of descriptive statistics are reported in Table 1. It includes mean values along with minimum and maximum values for every variable opted in this research. Moreover, the table reported standard deviation, kurtosis, skewness and Jarque–Bera test to check the normality of the variables. The mean value shows the average value of the variable during the period, while standard deviation shows the deviation of the values from their mean. The findings confirm that the average value for all variables is positive. The skewness and kurtosis show the normality of the data. In addition, the present study utilized the Jarque–Bera test to assert the normality in the chose factors. The discoveries of the JB test assert the expulsion of the null hypothesis at a 1% level of criticalness, which suggests that each factor is non-linear. The results further assert that there implies nonlinearity in each selected factor [42,68,69].

**Table 1.** Descriptive statistics analysis.

| Variable | EFP | GDP | GLO | NAR |
|---|---|---|---|---|
| Mean | 1.714 | 3053.656 | 51.470 | 1.709 |
| Minimum | 0.955 | 929.091 | 32.444 | 0.562 |
| Maximum | 2.644 | 6128.658 | 69.129 | 3.785 |
| Std. Dev. | 0.587 | 1633.931 | 13.230 | 0.811 |
| Skewness | 0.084 | 0.274 | −0.007 | 0.513 |
| Kurtosis | 1.448 | 1.768 | 1.391 | 2.373 |
| Jarque-Bera | 4.872 | 31.632 | 5.179 | 12.889 |
| Probability | 0.088 | 0.000 | 0.075 | 0.000 |

Source: authors' calculation.

There is an essential precondition of utilizing the ARDL bound testing methodology that the whole of the series of factors ought to be stationary at I(0) or I(1), nonetheless, not I(2). As appeared by Ouattara (2004), the disclosures of ARDL would be unacceptable if there is an I(2) factor presented in the studied model. Along these lines, it is vital to pick the stationarity of the dataset. Accordingly, the present investigation used two conventional unit root tests (for example, ADF and PP), and the results of the ADF and PP unit root are shown in Table 2. The outcomes showed that EFP, GDP, GLO and NAR demonstrate non-stationary conduct at a level and later changed into stationary at the first difference series. Moreover, the present investigation correspondingly utilized a basic structural break unit root test, for example [70], which imitates interruptions as clarified by [71]. Thinking about the issue of the break in the time plan, utilizing [70], the investigation additionally observed that all of the variables are stationary at I(1) as reported in Table 3. Along these

lines, it is confirmed that the present examination is utilized the ARDL technique as all the selected variables are not I(2).

**Table 2.** Unit Root Test Analysis.

| Variables | Unit Root Test (ADF) | | | | Unit Root Test (PP) | | | |
| | I(0) | | I(1) | | I(0) | | I(1) | |
| | C | C and T | C | C&T | C | C&T | C | C&T |
|---|---|---|---|---|---|---|---|---|
| **EFP** | 0.410 | 0.367 | −5.037 *** | −4.702 *** | 0.366 | 0.375 | −5.295 *** | −4.988 *** |
| **GDP** | −0.201 | −0.183 | −3.872 *** | −4.091 *** | −0.213 | −0.237 | −4.007 *** | −3.780 *** |
| **GLO** | −0.774 | −0.734 | −3.263 *** | −3.531 *** | −0.705 | −0.744 | −3.215 *** | −3.030 *** |
| **NAR** | 0.546 | 0.410 | −4.289 *** | −4.363 *** | 0.261 | 0.283 | −4.994 *** | −5.088 *** |

Note: EFP represents the ecological footprint, GDP describes the per capita of gross domestic product, GLO explains the globalization index including social, political and economic globalization, and NAR represents the rents for natural resources. Moreover *** refer to the level of significance at 1%. Source: Authors' calculation.

**Table 3.** Unit root test on Zivot–Andrews trended structural break.

| Variable | Level | | 1st Difference | |
| | T- Stat. | Time Break | T- Stat. | Time Break |
|---|---|---|---|---|
| EFP | −0.907 (1) | 2007 | −6.554 (1) *** | 1997 |
| GDP | −0.414 (1) | 2015 | −6.577 (1) *** | 1984 |
| GLO | 0.845 (1) | 2001 | −6.238 (1) *** | 1999 |
| NAR | −1.506 (1) | 2013 | −9.063 (1) *** | 2010 |

Note: parenthesis refer to lag order. *** refer to significance at 1% level. Source: authors' calculation.

Moreover, Reference [72] expressed that long-term affiliations concentrated on the best lag, and [73] likewise affirmed that utilizing an extra number of lags or taking a fewer lag could lose the most extreme imperious evidence of the model or might reason one-sided or biased estimations. Hence, sighted the status of perfect lags, the present investigation just 1 lag following the Schwarz info criteria (SIC). The discoveries of bound testing and nonlinear estimations are shown in Table 4. The outcome of F-statistics is greater than the tabulated values, which guarantees nonlinear long-term association among EFP, GDP, GLO and NAR in Thailand. Considering all the facts, the present examination pushes ahead to assess nonlinear ARDL coefficients.

**Table 4.** Bond test co-integration results.

| Model | F-Stat. | Up. Bond | Low. Bond |
|---|---|---|---|
| ln EFP/(ln GDP, ln GDP$^2$, ln GLO_POS, ln GLO_NEG, ln NAR_POS, ln NAR_NEG) | 64.583 | | |
| **Critical Values** | | | |
| 0.10 | | 4.50 | 1.70 |
| 0.05 | | 5.40 | 2.20 |
| 0.01 | | 6.90 | 2.80 |

Source: authors' calculation. Note: $p = o^+ = o^- = 0$, refer to combine null of no long-run relationship. The critical values are based on Narayan (2005).

After affirming the noteworthy nonlinear connection between EFP, GDP, GDP2, GLO and NAR in Thailand's economy, the present examination will continue towards long-run coefficients of our studied factors. The outcomes of long-run coefficients are reported in Table 5. The discoveries of NARDL affirmed that all factors altogether significant on the ecological footprint in Thailand. The outcomes further proposed that nonlinear and asymmetric association is found among EFP, GDP, GDP2, GLO and NAR in Thailand's economy. The outcomes also suggested that economic growth and squared of economic growth significantly impact the ecological footprint in Thailand. The results further sug-

gested that due to the negative shocks of globalization, the ecological footprint is increased by 28.5%; however, due to the positive shocks of GLO, the EFP has also been increased by 10.3%. The trend of both the shocks is significant and positive; however, the magnitudes of both shocks are significantly different from each other, which suggested a nonlinear association between GLO and EFP in Thailand. On the other hand, the effect of NAR on the EFP is significant and positive. The negative shocks of NAR increase the EFP by 24.3%; however, the positive shocks of NAR increase the EFP by 40.3%. In this case, the signs of both shocks are positive, but again the sizes of coefficients are significantly different from each other, suggesting a presence of nonlinear connection between NAR and EFP in Thailand's economy.

**Table 5.** NARDL Approach for long-run asymmetric.

| Variables | Coeff. | t-Stats | Prob. |
|-----------|--------|---------|-------|
| ln GDP | 0.375 | 4.092 | 0.000 |
| ln GDP$^2$ | −0.185 | 2.095 | 0.049 |
| ln GLO_NEG | 0.285 | 3.483 | 0.000 |
| ln GLO_POS | 0.103 | 4.094 | 0.000 |
| ln NAR_NEG | 0.243 | 3.968 | 0.000 |
| ln NAR_POS | 0.403 | 4.572 | 0.000 |

Dependent variable: ecological footprint. Source: authors' calculation.

This means that both globalization and natural resources are sources to increase the ecological footprint in Thailand. These findings are very rationale and justifiable as the consumption of natural resources and globalization, which mostly involved trade, increases the demand for natural resources, ultimately increasing the ecological footprint in a country. Moreover, the current study utilized nonlinear ARDL approach to test the environmental Kuznets curve in Thailand. The results suggested that economic growth is positive and significant; however, the square of economic growth is negative and significant, offering an inverted U-shape curve in Thailand. The results further confirm that, initially, the selected variables increase the ecological footprint, but after reaching a certain point, they started reducing the level of ecological footprint in Thailand.

Next, the discoveries of the diagnostic statistics of the NARDL technique are represented in Table 6. At this point, the criticalness estimation of LM and Breusch–Pagan–Godfrey are more noticeable than 0.100, which declares that the model is free from heteroscedasticity and serial correlation issues. Besides, the present examination has uncovered the *p*-estimation of the Ramsay RESET test, which is similarly more than 0.100, recommending that the current framework is sensibly specified. Finally, the present investigation points out the VIF estimation, which is 6.953, recommending no multicollinearity issue in the study's model.

**Table 6.** Diagnostic tests analysis.

| Diagnostic Test | Problem | *p*-Value | Status |
|-----------------|---------|-----------|--------|
| LM test | Serial Corr. | 0.192 | No Issue |
| BPagan-Godfrey | Hetero. | 0.402 | No Issue |
| Ramsey RESET test | Specification Err | 0.731 | No Issue |
| VIF | Multicoll. | 6.953 | No Issue |

Source: authors' calculation.

In the final phase, the present research utilized the asymmetric Granger causality introduced by [74]. The present research has opted for asymmetric causality to investigate the causal connection between the positive and negative shocks globalization, natural resources and ecological footprint in Thailand's economy. The outcomes are shown in Table 7. The findings of asymmetric Granger causality confirm that negative shocks of GLO and EFP have a significant causal relationship with the negative shocks of GLO and EFP

where the causality is running from the negative shocks of both variables to the negative shocks of other variables. In simple words, the findings confirm a significant bi-directional causal relationship between negative shocks of globalization and ecological footprint. On the other hand, the discoveries of asymmetric causality confirm that positive and negative shocks of NAR have a significant causal connection to the positive and negative shocks of EFP. However, the present study does not find any causal connection between positive and negative shocks of EFP to the positive and negative shocks of NAR in Thailand's economy.

**Table 7.** Asymmetric Granger causality analysis.

| Null Hypothesis | Wald Test | Bstrap 1% | Bstrap 5% | Bstrap 10% |
|---|---|---|---|---|
| GLO^− does not Granger cause EFP^− | 78.382 ** | 81.926 | 61.203 | 39.635 |
| GLO^− does not Granger cause EFP^+ | 4.5832 | 45.160 | 35.205 | 25.948 |
| GLO^+ does not Granger cause EFP^− | 26.782 | 57.207 | 44.255 | 28.652 |
| GLO^+ does not Granger cause EFP^+ | 59.391 | 130.870 | 103.732 | 89.953 |
| EFP^− does not Granger cause GLO^− | 38.582 ** | 40.948 | 32.148 | 24.452 |
| EFP^− does not Granger cause GLO^+ | 41.582 | 77.102 | 63.682 | 50.538 |
| EFP^+ does not Granger cause GLO^− | 29.879 | 51.989 | 39.880 | 31.312 |
| EFP^+ does not Granger cause GLO^− | 18.116 | 50.685 | 41.546 | 32.404 |
| NAR^− does not Granger cause EFP^− | 127.520 *** | 82.963 | 60.801 | 45.634 |
| NAR^− does not Granger cause EFP^+ | 59.727 ** | 103.192 | 53.748 | 39.294 |
| NAR^+ does not Granger cause EFP^− | 177.547 *** | 125.553 | 95.874 | 84.549 |
| NAR^+ does not Granger cause EFP^+ | 295.091 *** | 89.970 | 72.123 | 56.408 |
| EFP^− does not Granger cause NAR^− | 38.143 | 100.058 | 63.889 | 51.070 |
| EFP^− does not Granger cause NAR^+ | 33.085 | 78.674 | 50.538 | 41.747 |
| EFP^+ does not Granger cause NAR^− | 1.447 | 39.837 | 24.397 | 7.900 |
| EFP^+ does not Granger cause NAR^− | 23.247 | 104.258 | 77.950 | 53.434 |

Note: ** and *** indicate statistical significance at 5% and 1% level, respectively. Critical values are obtained from 10,000 bootstrap replications. Source: authors' calculation.

## 4. Discussion and Conclusions

The present study investigated the asymmetric impact of natural resources and globalization on ecological footprint in the presence of EKC in Thailand. The study used annual time series data from 1970 to 2018. The findings confirm that the effect of globalization and natural resources are significant and nonlinear. These results align with Figge et al. [6], who also exposed that globalization and natural resources significantly affect the ecological footprint. However, the effect of negative shocks of globalization and natural resources is more dominant on the ecological footprint in Thailand than positive shocks of both variables. These results are also the same as Schandl et al. [28], who also examined that natural resources and globalization positively associate with the ecological footprint. Moreover, the present study has also tested the presence of EKC in Thailand, and the findings confirm the presence of an inverted U-shape curve in Thailand's economy. These results are also similar to Destek et al. [15], who also found the presence of an inverted U-shape curve in newly industrialized countries. On the other hand, the findings of asymmetric Granger causality confirm a bi-directional causal connection from negative shocks of globalization (ecological footprint) to the negative shocks of ecological footprint (globalization). This outcome is matched with the outcome of Charfeddine et al. [12] that also found bi-directional causality among globalization and ecological footprint. Moreover, the findings further suggested a unidirectional causal connection between natural resources and ecological footprint where causality runs from the positive and negative shocks of natural resources to the positive and negative shocks of ecological footprint. These outcomes are also in line with the output of Hassan et al. [3] that also found unidirectional causal relation among natural resources and ecological footprint.

Thus, the present study has concluded that the high level of natural resources and increasing level of globalization put a significant role on the ecological footprint in Thailand. In addition, as much as the natural resources hold by the country, the people of the

country consume the resources faster and generate wastage. Moreover, globalization also forces people to use extra-ordinary resources to survive in the global market. Thus, this study suggested to the regulators that they should develop effective policies related to the effective usage of natural resources and positively respond to globalization affecting the environment.

**Author Contributions:** Conceptualization, H.I.H. and M.H.; methodology, H.I.H.; software, M.H.; validation, M.H., Z.D.-P. and K.S.-W.; formal analysis, F.K. and M.H.; investigation, Z.D.-P.; resources, K.S.-W.; data curation, K.S.-W.; writing—original draft preparation, H.I.H.; writing—review and editing, F.K.; visualization, Z.D.-P.; supervision, F.K.; project administration, H.I.H.; funding acquisition, F.K. All authors have read and agreed to the published version of the manuscript.

**Funding:** This research is funded by publication fund using PTJ code 12051, project code 9001103 under Universiti Putra Malaysia (UPM) and the project is also funded under the program of the Minister of Science and Higher Education titled "Regional Initiative of Excellence" in 2019–2022, project number 018/RID/2018/19, the amount of funding PLN 10 788 423,16.

**Institutional Review Board Statement:** Not applicable.

**Informed Consent Statement:** Not applicable.

**Data Availability Statement:** Data available in a publicly accessible repository that does not issue DOIs Publicly available datasets were analyzed in this study. This data can be found here: https://kof.ethz.ch/en/forecasts-and-indicators/indicators/kof-globalisation-index.html, assessed on 1 January 2021; https://databank.worldbank.org/source/world-development-indicators, assessed on 1 January 2021.

**Acknowledgments:** We would like to thank the editors and the anonymous referees of the journal for constructive comments and suggestions, which have significantly helped to improve the contents of the paper. The usual caveats apply.

**Conflicts of Interest:** The authors declare no conflict of interest.

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
