# Peer review of "The Role of Globalization, Economic Growth and Natural Resources on the Ecological Footprint in Thailand: Evidence from Nonlinear Causal Estimations"

_processes, doi:10.3390/pr9071103_

Round 1
Reviewer 1 Report
Authors proposed the following paper entitled “The Role of Globalization, Economic Growth and Natural Re-2 sources on the Ecological Footprint in Thailand: Evidence from Nonlinear Causal Estimations”
The work is interesting and the issue proposed here has a worldwide relevance. Almost 50 years of registered data were used in this work.
English is quite good, even if a careful revision of the use of English should be needed.
here is the list of some observations and comments to the paper.
Line 36. I would not be so totalitaristic as “responsible for this destruction”. maybe it could be better “responsible for this progressive worsening of living conditions”.
Line 39. double space in this line needs to be removed.
Line 40. “ is a global challenge [1]. [2] whilst using”. there is a problem here with the references and the dot among them.
I suggest adding reference at the end of the sentence in Line 51.
Line 69. “an explanation of why it is so they borrowed the idea” I suggest checking the English here.
Line 73. “there are studies like [9] which”. I suggest modifying such as “Surname et al [9] studied/discovered that…”
Line 78. “as well negative”. maybe better “as well as…”
Line 86. MENA. which region is this? please add brief explanation for foreign readers.
Line 88 “used facts of [13] and”. maybe better “used facts cited in the work by Surname et. [13”
Line 89. a double space is reported in this line.
Lines 95-96. this period seems to be too long.
line 105. could you please define gha unit of measure?
Line 140. “with the passage of time”. maybe here there should be another expression such as “over time” or something equivalent.
Line 175. please define GDP
Line 177. “symmetry“. maybe better “balance”?
I suggest the addition of an abbreviation list according to this journal guidelines, due to the high number of acronyms employed in this manuscript.
From Lines 295 to 300, it is not clear if authors are presenting the aim of their study or it is the description of the reference 42 content. please clarify in the text before moving to following sections.
Indeed, the aims are clearly presented at the beginning of the material and method section. In my opinion, aims should be moved to the end of the introduction section, and should not be here.
Equations 1 and 2 contain parameters that should be defined.
Lines 372-378, maybe not necessary to say how much is the estimation, if everything is reported in the following table. it could be better to add more comments on the table.
Additional comment
I suggest adding at least one figure.
Author Response
"Please see the attachment."

Reviewer 2 Report
Dear Authors,
The publication submitted for evaluation, The Role of Globalization, Economic Growth and Natural Resources on the Ecological Footprint in Thailand: Evidence from Nonlinear Causal Estimations by, Hafezali Iqbal Hussain, Muhammad Haseeb, Fakarudin Kamarudin, Zdzisława Dacko-Pikiewicz and Katarzyna Szczepańska-Woszczyna takes up, and presents issue of impact of globalization, economic growth and natural resources on ecological footprint in Thailand. The presented work, apart from its cognitive value, is of a great practical significance for further climate change mitigation, and adaptation activities.
I appreciate your efforts which resulted in conducting such an interesting study. But, in order to be published in a prestigious journal your paper needs some improvements. Please consider the following comments:
- the Results must be discussed within the context of previous researches, so I recommend you to try to improve this section also. There are currently no references to literature.
Author Response
"Please see the attachment."

Reviewer 3 Report
Thank you for submitting to Processes. While I found the paper interesting, I believe that it is not suitable for publication in its current form, needing major revision. I have a number of concerns.
Firstly, the work needs significant grammatical work, and assumes that readers are familiar with many terms and acronyms that are never defined. The first sentence of the abstract immediately raises flags regarding the quality of the submission. Apart from this, my major concerns are the following:
- No clear motivation and novelty demonstrated for the current work. While the introduction gives an overly elaborate background, it never clearly and concisely states what the gap in the literature is and what the paper does to address this gap.
- While the work gives many numbers and figures, the results are never truly explained, nor are any figures presented that may give readers insight into the results. Apart from the statistical testing and overall patterns, the work should go further in the results to posit reasons for the results and to highlight anything unusual or similar in relation to previous works and results.
- The source of the data upon which the study is based is never given
- The conclusion does not conclude.
Apart from these major concerns, I have the following comments that arose during reading:
ABSTRACT
ARDL not defined here. Everything is written in the past tense. The abstract does not give the motivation for the work, nor on the meaning of the results.
Introduction
MENA not defined.
P3. Lines 133-134: “strong and robust understanding” – this is not clear.
Clearly lumping inappropriate self-citations in the reference lump on p. 3. Lines 137.
FDI not defined
Page 5 seems to be an overly-exhaustive list of all the studies done. It seems to me that this does not enhance motivation or highlight the novelty of the work. I suggest summarising the key patterns and outputs.
KOF index not defined.
Introduction does not link to the section 2. Please add a paragraph at the end of 1 to summarise the gap in the literature. It is unclear from the introduction, where the novelty of the current work lies.
Section 2.
GLO and EFP are not defined where they are first referred to.
Page 7, line 314: More self-citations that seem inappropriate
Page 7, line 329: GDP missing square
SIC not defined when first appears.
- Results
Where did the data come from that the tests and regressions were applied to?
I feel strongly that there needs to be a visualisation of the curves and relationships to inform the readers and to provide evidence of the relationships
The write-up does not give any real insights into why the model behaves in this way, nor what is novel or interesting about the results in relation to previous work.
Discussion and conclusion does not truly conclude, or provide any feeling for the novelty of the work or analysis of what the results infer.
Author Response
"Please see the attachment."

Round 2
Reviewer 1 Report
Authors provided a new version of their paper, adding more information where requested and analyzing many aspects that before were not directly explained. The use of English is very good; the paper deserves now to be published.
I only have to ask a few minor revisions, such as:
“examined frequently” is a good way to monitor and study the issue of environmental pollution however, it could be important to talk about sustainability, that encounters the environmental problems also in terms of social and economic impact.
Line 71. Used by. I suggest adding the first author of this study, like it was correctly performed in line74.
line236. how these 16 european economics were chosen to support this study?
Line 238. 0.81 %, not percent.
line 525. “the people are faster to consume the resources”. Maybe I suggest saying: “people consume resources faster”.
There is not an abbreviation list, but only the description of the variables included in formulas.
Reviewer 3 Report
Thank you to the authors for addressing the majority of my concerns. I believe the paper to be significantly improved. The quality of the writing (English language and style) still needs improvement.
